# The Impact Once-Weekly Semaglutide 2.4 mg Will Have on Clinical Practice: A Focus on the STEP Trials

**DOI:** 10.3390/nu14112217

**Published:** 2022-05-26

**Authors:** Khaled Alabduljabbar, Werd Al-Najim, Carel W. le Roux

**Affiliations:** 1Department of Family Medicine and Polyclinics, King Faisal Specialist Hospital and Research Centre, Riyadh 11211, Saudi Arabia; khalabduljabbar@kfshrc.edu.sa; 2Diabetes Complications Research Centre, Conway Institute, University College Dublin, D04 V1W8 Dublin, Ireland; werd.al-najim@ucd.ie

**Keywords:** semaglutide, obesity, STEP program, weight loss, weight management, clinical trial, GLP-1

## Abstract

Obesity is a complex and chronic disease that raises the risk of various complications. Substantial reduction in body weight improves these risk factors. Lifestyle changes, including physical activity, reduced caloric ingestion, and behavioral therapy, have been the principal pillars in the management of obesity. In recent years, pharmacologic interventions have improved remarkably. The Semaglutide Treatment Effect in People with Obesity (STEP) program is a collection of phase-III trials geared toward exploring the utility of once-weekly 2.4 mg semaglutide administered subcutaneously as a pharmacologic agent for patients with obesity. All the STEP studies included diet and exercise interventions but at different intensities. This review paper aims to explore the impact of the behavioral programs on the effect of semaglutide 2.4 mg on weight loss. The results of the STEP trials supported the efficacy of high-dose, once-weekly 2.4 mg semaglutide on body weight reduction among patients with obesity with/without diabetes mellitus. Semaglutide was associated with more gastrointestinal-related side effects compared to placebo but was generally safe and well tolerated. In all the STEP studies, despite the varying intestines of the behavioral programs, weight loss was very similar. For the first time, there may be a suggestion that these behavioral programs might not increase weight reduction beyond the effect of semaglutide. Nevertheless, the importance of nutritional support during substantial weight loss with pharmacotherapy needs to be re-evaluated.

## 1. Introduction

Obesity is a complex and chronic disease and has a wide array of complications, including hypertension, hypercholesteremia, type 2 diabetes, cardiovascular disease, and some cancers [1,2,3,4,5,6]. Lifestyle interventions, comprising physical activity, reduced caloric ingestion, and behavioral therapy, have been the principal pillars in the management of obesity supported by pharmacotherapy and bariatric surgery [7,8,9]. However, weight loss maintenance has remained challenging [10]. Pharmacotherapy is usually used for individuals with a body mass index (BMI) ≥30 kg/m^2^ or ≥27 kg/m^2^ with ≥1 coexisting obesity complication [7,8,9], but the cost, efficacy, and tolerability curbs its utilization [11].

Only a few obesity medications have received approval by the US Food and Drug Administration (FDA), namely naltrexone-bupropion combination, phentermine-topiramate combination, orlistat, setmelanotide, liraglutide, and semaglutide [12,13,14]. These medications, except phentermine-topiramate, are also approved by the European Medicines Agency (EMA) to be used in Europe [15,16]. The mechanism of actions and the approval status for these medications are presented in Table 1.

Semaglutide belongs to the family of glucagon-like peptide-1 analogs. Mechanistically, semaglutide is an incretin, which blocks glucagon release, postpones gastric clearing, reduces energy intake, stimulates satiety, and reduces hunger and appetite via peripheral and central nervous system actions [17]. Semaglutide was initially approved for the management of type 2 diabetes mellitus [18]. The observation that the GLP-1 analogs reduce body weight prompted the exploration of this class of medications as drugs to treat obesity [19,20,21].

The Semaglutide Treatment Effect in People with Obesity (STEP) program is a collection of 15 multi-institutional, phase-III, randomized, double-blind, placebo-controlled trials geared toward the authorization of once-weekly 2.4 mg semaglutide administered subcutaneously as an obesity medication. Each trial was designed to investigate the efficacy and safety of 2.4 mg semaglutide in people with overweight or obesity, taking in consideration patients’ ethnicities, certain comorbidities, different age groups, or the parallel control arm interventions. Six of the program trials (STEP 1–4, 6, and 8) were published; the STEP 5 trial has been completed but not yet published, and the remining trials, including STEP 7, have not been completed yet. Herein, we document a narrative review focused on the clinical summary of the STEP trials, highlight limitations, and outline future directions, with a specific focus on the potential future role of lifestyle changes in obesity management involving such effective medications.

## 2. The STEP 1 Trial

The STEP 1 trial (ClinicalTrials.gov identifier: NCT03548935) included adults with obesity or overweight (BMI ≥ 27 kg/m^2^) with at least one obesity complication [22]. Major exclusion criteria included diabetes mellitus, HbA1c ≥ 6.5%, or the use of anti-obesity medications in the past 12 weeks. In a 2:1 ratio, the trial randomized 1961 adults to either semaglutide or placebo. Semaglutide was administered in a dose-escalated fashion: the initial once-weekly dose of 0.25 mg was sustained for four weeks; the dose was then titrated to 0.5 mg, 1 mg, 1.7 mg, and 2.4 mg every four weeks. The 2.4 mg once-weekly dose was then maintained for 54 weeks. Overall, the duration of the study was 75 weeks; the treatment (semaglutide or placebo) lasted for 68 weeks, trailed by a follow-up interval of 7 weeks with no medication. The protocol included an unsupervised lifestyle intervention administered to all participants, consisting of a daily 500 kcal deficit diet and weekly 150 min of physical activity. The average age and BMI of the participants were 46 years and 37.9 kg/m^2^, respectively. The majority of the participants were females (74.1%) and of White ethnicity (75.1%). Less than half of the participants had prediabetes (43.7%).

Semaglutide with lifestyle intervention resulted in more weight loss over 68 weeks compared to placebo with lifestyle intervention (mean difference (MD) = –12.4%, 95% confidence interval (CI): −13.4, −11.5). Moreover, the proportions of participants treated with semaglutide achieving ≥5%, ≥10%, and ≥15% weight loss at week 68 were 86.4%, 69.1%, and 50.5%, respectively. In addition, the semaglutide arm had substantial improvements in various anthropometric (BMI and waist circumference), inflammatory (C-reactive protein), blood pressure (diastolic and systolic), glycemic (HbA1c and fasting plasma glucose), and lipid (total cholesterol, triglycerides, and low-density lipoprotein cholesterol) parameters in contrast to the placebo arm. Semaglutide also substantially improved physical function scores compared to the placebo assessed by the 36-item Short Form Health Survey (SF-36) and the Impact of Weight on Quality of Life–Lite Clinical Trials Version (IWQOL-Lite-CT) questionnaire.

The rate of any reported side effect was higher with semaglutide contrasted with placebo (89.7% vs. 86.4%, respectively). The number of reported serious side effects was greater in the semaglutide arm compared to the placebo arm. The rate of drug termination was also greater in the semaglutide arm (7.0% vs. 3.1%), mostly due to gastrointestinal-related symptoms (4.5% vs. 0.8%). Gallbladder-related symptoms occurred in 2.6% of patients in the semaglutide arm and 1.2% in the placebo arm. The most commonly documented side effects in ≥10% of the semaglutide vs. placebo patients were nausea (44.2% vs. 17.4%), diarrhea (31.5% vs. 15.9%), vomiting (24.8% vs. 6.6%), constipation (23.4% vs. 9.5%), and nasopharyngitis (21.5% vs. 20.3%). The rates of hypoglycemia, acute pancreatitis, and injection site reactions were infrequent in the participants who received semaglutide (0.6%, 0.2%, and 5%, respectively).

In summary, among patients with BMI ≥ 27 kg/m^2^, the STEP 1 trial concluded that once-weekly 2.4 mg semaglutide plus usual lifestyle adjustment was more beneficial than lifestyle interventions alone in reducing body weight and other cardiometabolic risk factors.

## 3. The STEP 2 Trial

The STEP 2 trial (ClinicalTrials.gov identifier: NCT03552757) included adults with BMI ≥ 27 kg/m^2^ and HbA1c ranging from 7% to 10%; all participants were diagnosed with type 2 diabetes mellitus ≥6 months prior to study screening [23]. In a 1:1:1 ratio, the trial randomized 1210 participants to 2.4 mg semaglutide, 1.0 mg semaglutide, or placebo. All the participants in this study had the same lifestyle intervention as the STEP 1 trial. Semaglutide was administered in a dose-escalated fashion until reaching the targeted maintenance doses. Overall, the study duration was 75 weeks; the treatment (semaglutide or placebo) lasted for 68 weeks, trailed by a follow-up period of 7 weeks with no medication. Participants’ average age and BMI were 55 years and 35.7 kg/m^2^, respectively. The average HbA1c and interval of type 2 diabetes mellitus were 8.1% and 8 years. Slightly more than half of the participants were females (50.9%) and of White ethnicity (62.1%).

The 2.4 mg semaglutide with lifestyle intervention reduced body weight more than placebo and lifestyle intervention during the 68 weeks (MD = –6.2%, 95% CI: −7.3, −5.2). Moreover, the proportions of participants who had ≥5%, ≥10%, and ≥15% weight loss at week 68 were 68.8%, 45.6%, and 25.8%, respectively. Semaglutide also improved systolic blood pressure, HbA1c, waist circumference, and physical function scores. In addition, the analysis of exploratory secondary endpoints revealed beneficial reductions in lipid (triglycerides, very-low-density lipoprotein cholesterol, and free fatty acids), glycemic (HbA1c, fasting serum insulin, and fasting plasma glucose), inflammatory (C-reactive protein), and blood pressure (diastolic) profiles in support of the semaglutide 2.4 mg arm compared to the placebo arm.

The rate of any reported side effect was greater in the 2.4 mg semaglutide arm contrasted with the placebo arm (87.6% vs. 76.9%). Moreover, the number of reported serious side effects was comparable between both treatment arms. Additionally, the rate of drug termination was higher in the semaglutide 2.4 mg arm (6.2% vs. 3.5%), mostly secondary to gastrointestinal-related symptoms (4.2% vs. 1.0%). Gallbladder-related symptoms occurred in only 0.2% and 0.7% of the semaglutide 2.4 mg and placebo arms, respectively. In contrast with the placebo arm, the most commonly documented side effects in ≥10% of the semaglutide 2.4 mg patients included nausea (33.7% vs. 9.2%), diarrhea (21.3% vs. 11.9%), vomiting (21.8% vs. 2.7%), constipation (17.4% vs. 5.5%), and nasopharyngitis (16.9% vs. 14.7%). The rates of hypoglycemia, acute pancreatitis, and injection site reactions were infrequent in the semaglutide 2.4 mg arm (5.7%, 0.2%, and 3.0%, respectively).

In summary, among adults with type 2 diabetes mellitus and BMI ≥ 27 kg/m^2^, the STEP 2 trial concluded that once-weekly 2.4 mg semaglutide plus lifestyle modification was better than lifestyle modification alone for weight loss and other cardiometabolic risk factors.

## 4. The STEP 3 Trial

The STEP 3 trial (ClinicalTrials.gov identifier: NCT03611582) included adults with the same inclusion and exclusion criteria as the STEP 1 trial [22,24]. In a 1:1 ratio, the trial randomized 611 participants to either semaglutide combined with very intensive behavior therapy or placebo with very intensive behavior therapy. The intensive behavior therapy was the major difference between STEP 3 compared to STEP 1 and STEP 2 trials [22,23]. It comprised a low-calorie diet during the opening 8 weeks, in addition to concentrated behavioral therapy sessions and physical exercise during the 68 weeks.

The participants were provided with a low-calorie diet (1000–1200 kcal/day) served as meal replacements for the first 8 weeks. Then, they were gradually transferred to hypo-caloric diet (1200–1800 kcal/day) of conventional food for the remainder of the trial. After eight weeks, the calorie intake was calculated based on randomization body weight unless the participant’s BMI reached ≤22.5 kg/m^2^. The recommended caloric intake was then re-calculated with no energy deficit until the end of the trial.

Physical activity was prescribed from randomization and was tailored to achieve a goal of 100 min of physical activity/week. Participants were counseled to incorporate moderate-intensity activities within the exercises and were requested to increase their weekly physical activity target by 25 min every four weeks to reach 200 min/week. Furthermore, a total of 30 counseling sessions of intensive behavioral therapy were provided over the 68 weeks, covering various topics related to dietary changes, physical activities, and behavioral strategies to ensure the appropriate implementation and compliance with the intervention.

Similar to the STEP 1 and STEP 2 trials [22,23], semaglutide was administered in a dose-escalated fashion until reaching the targeted maintenance dose. The duration of the study was 75 weeks; the treatment (semaglutide with very intensive behavior therapy or placebo with very intensive behavior therapy) lasted for 68 weeks, trailed by a follow-up period of 7 weeks with no medication. The average age and BMI of participants were 46 years and 38 kg/m^2^, respectively. The majority of the research participants were females (81.0%) and of White ethnicity (76.1%).

Semaglutide with very intensive behavior therapy resulted in a more significant weight loss from baseline to week 68 compared to placebo with very intensive behavior therapy (MD = –10.3%, 95% CI: −12.0, −8.6). Moreover, the proportions of research participants who achieved ≥5%, ≥10%, and ≥15% body weight loss at week 68 were 86.6%, 75.3%, and 55.8%, respectively. The placebo with intensive behavior therapy was also effective, albeit less so than semaglutide with intensive behavior, in causing ≥5%, ≥10%, and ≥15% weight loss at 68 weeks in 46.7%, 27.0%, and 13.2% of patients, respectively. Semaglutide had significant improvements in systolic blood pressure and waist circumference but was not different in the physical functioning scores compared to the placebo arm. Similar to the previous STEP trials, the analysis of exploratory secondary endpoints revealed beneficial reductions in various lipid, glycemic, inflammatory, and blood pressure (diastolic) profiles in support of the semaglutide arm compared to the placebo arm.

The rate of ≥1 reported side effect was comparable between the semaglutide and placebo arms (95.8% vs. 96.1%). Moreover, the number of reported serious side effects was greater in the semaglutide arm contrasted with the placebo arm (9.1% vs. 2.9%). Additionally, the rate of drug termination was higher with semaglutide (5.9% vs. 2.9%), mostly secondary to gastrointestinal-related symptoms (3.4% vs. 0.0%). Gallbladder-related symptoms took place in only 4.9% and 1.5% of the semaglutide and placebo arms, respectively. In contrast with the placebo arm, the most common side effects in ≥10% of the semaglutide arm included nausea (58.2% vs. 22.1%), constipation (36.9% vs. 24.5%), diarrhoea (36.1% vs. 22.1%), vomiting (27.3% vs. 10.8%), and nasopharyngitis (22.1% vs. 24.0%). The rates of hypoglycemia, acute pancreatitis, and injection site reactions were infrequent among individuals who received the semaglutide therapy (0.5%, 0%, and 5.4%, respectively).

In summary, the STEP 3 trial concluded that semaglutide plus intensive behavior therapy, including an initial low-calorie intake and rigorous behavioral therapy, culminated in clinically meaningful improvements in body weight and other cardiometabolic risk factors compared with the placebo treatment.

## 5. The STEP 4 Trial

The STEP 4 trial (ClinicalTrials.gov identifier: NCT03548987) [25] included adults with the same inclusion and exclusion criteria as the STEP 1 and STEP 3 trials [22,24]. For all research participants (*n* = 902), semaglutide was administered in a dose-escalated fashion for a 20-week run-in period (16 weeks of dose intensification starting with 0.25 mg until reaching 2.4 mg, trailed by 4 weeks of maintenance dose 2.4 mg). Only patients who were able to tolerate semaglutide 2.4 mg were included in the randomization period, thus excluding those who could not achieve the top dose of the medication. Overall, the duration of the study was 75 weeks; the run-in period lasted for 20 weeks, trailed by a randomization in 2:1 ratio (*n* = 803) to either 2.4 mg semaglutide or placebo and followed by a follow-up period of 7 weeks with no medication. The lifestyle intervention was a 500 kcal deficit diet and 150 min of exercise per week, similar to STEP 1, but less intensive than STEP 3 [22,24]. The average age and BMI of participants were 46 years and 38.4 kg/m^2^, respectively. The majority of the research participants were females (79.0%) and of White ethnicity (83.7%).

After the 20-week run-in interval, the average weight loss was 10.6%, and several improvements were witnessed in blood pressure (systolic and diastolic), HbA1c, lipid parameters, and waist circumference. Between week 20 to week 68, participants randomized to ongoing semaglutide continued to lose weight as opposed to those randomized to the placebo arm who gained weight during the same period (MD = –14.8%, 95% CI: −16.0, −13.5). Moreover, the continued semaglutide arm achieved significant decreases in systolic blood pressure (MD = –3.9 mmHg, 95% CI: −5.8, −2.0) and waist circumference (MD = –9.7 cm, 95% CI: −10.9, −8.5). Furthermore, the physical function scores were significantly better with continued semaglutide. Similar, to the earlier STEP trials [22,23,24], the analysis of exploratory secondary endpoints revealed beneficial reductions in various lipid and glycemic profiles in support of the continued semaglutide arm contrasted with the switched placebo arm.

The rate of any adverse event was greater in the continued semaglutide arm than the switched placebo arm (81.3% vs. 75.0%). The number of serious side effects was higher in the continued semaglutide arm contrasted with the switched placebo arm (7.7% vs. 5.6%). However, the rate of drug termination was comparable between both arms (2.4% vs. 2.2%). Gastrointestinal and gallbladder-related symptoms took place in 41.9% and 2.8% of the continued semaglutide arm and in 26.1% and 3.7% of the switched placebo arm. In contrast with the switched placebo arm, the most commonly documented side effects in ≥5% of the continued semaglutide arm included diarrhoea (14.4% vs. 7.1%), nausea (14.0% vs. 4.9%), constipation (11.6% vs. 6.3%), nasopharyngitis (10.8% vs. 14.6%), and vomiting (10.3% vs. 3.0%). The rates of hypoglycemia, acute pancreatitis, and injection site reactions were infrequent in the continued semaglutide arm (0.6%, 0%, and 2.6%, respectively).

In summary, the STEP 4 trial concluded that once-weekly continued 2.4 mg semaglutide after a 20-week run-in interval plus standard lifestyle modifications led to sustained body weight loss over the next 48 weeks contrasted with individuals who switched to placebo who started regaining weight.

## 6. The STEP 5 Trial

The STEP 5 trial (ClinicalTrials.gov identifier: NCT03693430) included adults with the same inclusion and exclusion criteria as the STEP 1, STEP 3, and STEP 4 trials [22,24,25,26]. In a 1:1 ratio, the trial randomized 304 participants to either semaglutide with standard lifestyle modifications of a 500 kcal deficit diet and 150 min of exercise per week or placebo with standard lifestyle modifications of a 500 kcal deficit diet and 150 min of exercise per week. Semaglutide was administered in a dose-escalated fashion until reaching the targeted maintenance dose of 2.4 mg (end of week 20), which was continued until week 104. The study continued for 111 weeks; the treatment (semaglutide or placebo) lasted for 104 weeks, trailed by a follow-up period of 7 weeks with no medication. The average age and BMI of participants were 47 years and 38.5 kg/m^2^, respectively. The majority of the research participants were females (78.0%) and of White ethnicity (93.1%).

Semaglutide with standard lifestyle modifications achieved more weight loss from baseline to week 104 contrasted with placebo with standard lifestyle modifications (MD = –12.6%, 95% CI: −15.3, −9.8). Moreover, the proportions of research participants with ≥5%, ≥10%, ≥15%, and ≥20% weight loss at week 104 with semaglutide were 77.1%, 61.8%, 52.1%, and 36.1%, respectively. In addition, the semaglutide arm had significant improvements in various cardiovascular risk factors, such as systolic blood pressure (MD = –4.2 mmHg, 95% CI: –7.3, –1.0), diastolic blood pressure (MD = –3.7 mmHg, 95% CI: –6.1, –2.1), C-reactive protein (MD = –53.1%, 95% CI: –63.2, –40.0), and waist circumference (MD = –9.2 cm, 95% CI: –12.2, –6.2). The semaglutide arm also had significant improvements in various metabolic risk factors, such as HbA1c (MD = –0.33%, 95% CI: –0.41, –0.25), fasting plasma glucose (MD = –9.2 mg/dL, 95% CI: –12.0, –6.5), fasting serum insulin (MD = –27.4%, 95% CI: –39.3, –13.3), and triglycerides (MD = –22.0%, 95% CI: –29.8, –13.2).

The rate of any adverse event was greater after semaglutide compared to placebo (96.1% vs. 89.5%). The number of reported serious side effects was unexpectedly lower in the semaglutide arm contrasted with the placebo arm (7.9% vs. 11.8%). The rate of drug termination was similar in the semaglutide and placebo arms (5.9% vs. 4.6%). Gastrointestinal and gallbladder-related symptoms took place in 82.2% and 2.6% of the semaglutide arm. In contrast, gastrointestinal and gallbladder-related symptoms took place in 53.9% and 1.3% of the placebo arm. The rates of hypoglycemia, acute pancreatitis, and injection site reactions were rare in the semaglutide arm (2.6%, 0%, and 6.6%, respectively).

In summary, the STEP 5 trial concluded that once-weekly semaglutide dose (plus lifestyle modifications) led to sustained body after two years of treatment, improved cardiovascular and metabolic risk factors, and depicted satisfactory safety profile compared with placebo.

## 7. The STEP 6 Trial

The STEP 6 trial (ClinicalTrials.gov identifier: NCT03811574) included east Asian adults, with or without type 2 diabetes, who reported a failed weight loss dietary attempt and had a BMI ≥ 27 kg/m^2^ with two or more weight-related medical problems or a BMI ≥ 35 kg/m^2^ with at least one weight-related medical problem [27]. The main exclusion criteria were previous or planned anti-obesity treatment or surgery and bodyweight changes of 5 kg or more in the past 3 months before screening. The Asian ethnicity was the major difference between STEP 6 compared to the previous STEP trials (the majority were White) [22,23,24,25,26]. In a 4:1:2:1 ratio, the trial randomized 401 participants to either semaglutide 2.4 mg or placebo, or semaglutide 1.7 mg or placebo. The dose was administered in an escalated fashion until reaching the targeted doses. All the participants were advised to follow the standard lifestyle modifications similar to STEP 5 trial. Overall, the duration of the study was 75 weeks; the treatment lasted for 68 weeks, trailed by a follow-up interval of 7 weeks with no medication. The average age and BMI of participants were 51 years and 31.9 kg/m^2^, respectively. All the research participants were Asian (100%), and the majority were males (63%).

The semaglutide with lifestyle intervention in both doses (2.4 mg and 1.7 mg) reduced body weight more than placebo with lifestyle intervention during the 68 weeks (MD = −11.06%, 95% CI: −12.88, −9.24 and MD = −7.52%, 95% CI: −9.62, −5.43, respectively). Moreover, the proportions of participants who had ≥5%, ≥10%, and ≥15% weight loss at week 68 with semaglutide 2.4 mg were 83%, 61%, and 41%, respectively. The treatment arm also showed significant reductions in waist circumference, systolic blood pressure, and HbA1c. In addition, the analysis of exploratory secondary endpoints revealed favorable reductions among semaglutide groups in BMI, fasting plasma glucose, C-reactive protein, and plasminogen activator inhibitor-1, lipid profile (except for high-density lipoprotein cholesterol). An improvement in the physical function score was noted in the semaglutide 2.4 mg group. From baseline to week 68, greater reductions in abdominal visceral fat area were observed in the semaglutide 2.4 mg (−40%) and 1.7 mg (−22.2%) groups than the placebo group (−6.9%).

The rate of reported adverse events was 86% in the semaglutide 2.4 mg group, 82% in the semaglutide 1.7 mg group, and 79% in the placebo group. Unexpectedly, the percentage of serious adverse events was lower in the semaglutide 2.4 mg arm (5%) contrasted with semaglutide 1.7 mg and placebo arms (7% each). The rate of drug termination was higher in semaglutide groups (3%) compared to placebo (1%). Gallbladder-related symptoms took place in only 1% in all groups. Gastrointestinal-related symptoms, which were mostly mild to moderate, were more common in semaglutide 1.7 mg group (64%) than semaglutide 2.4 mg group (59%) or placebo group (30%). The rates of hypoglycemia and acute pancreatitis were 0% in all arms. Injection site reactions were reported in only four participants in the semaglutide 2.4 mg arm.

In summary, among east Asian patients with BMI ≥ 27 kg/m^2^, with or without type 2 diabetes, the STEP 6 trial concluded that once-weekly 2.4 mg semaglutide plus lifestyle adjustment led to significant reductions in body weight, abdominal visceral fat, and other cardiometabolic risk factors compared with placebo in this population.

## 8. The STEP 8 Trial

The STEP 8 trial (ClinicalTrials.gov identifier: NCT04074161) was an open label with treatment arms and double-blinded against matched placebo arms [28]. It included adults with the same inclusion and exclusion criteria as the STEP 1 trial. In a 3:1:3:1 ratio, the trial randomized 338 participants to either once-weekly semaglutide (dose-escalation to 2.4 mg over 16 weeks), or matching placebo, or once-daily liraglutide (dose escalation to 3.0 mg over 4 weeks), or matching placebo. Both semaglutide and liraglutide are long-acting GLP-1 analogs. As a result of the substitution of amino acids that prevents the degeneration of dipeptidyl peptidase 4 and addition of C18 fatty acids, semaglutide has a half-life of 165 h, whereas liraglutide’s half-life is about 13 h [26]. All the participants in this trial had the same lifestyle intervention as the STEP 1 trial. The study continued for 75 weeks; the treatments lasted for 68 weeks, trailed by a 7-week follow-up period with no medications. The mean age and BMI of participants were 49 years and 37.5 kg/m^2^, respectively. The majority of the research participants were females (78.4%) and of White ethnicity (73.7%).

Semaglutide with lifestyle modifications resulted in a more significant weight loss from baseline to week 68 compared to liraglutide with lifestyle modifications (MD = −9.4%, 95% CI: −12.0, −6.8). Furthermore, the proportions of the semaglutide patients who achieved ≥5%, ≥10%, ≥15%, and ≥20% body weight loss at week 68 were 87.2%, 70.9%, 55.6%, and 38.5%, respectively. Liraglutide was also effective, albeit less than semaglutide, in causing ≥5%, ≥10%, ≥15%, and ≥20% weight loss at 68 weeks in 58.1%, 25.6%, 12%, and 6% of participants, respectively. At week 68, reductions in BMI, waist circumference, blood pressure, HbA1c, fasting plasma glucose, triglyceride, total cholesterol, very-low-density lipoprotein cholesterol, and C-reactive protein levels were significantly greater with semaglutide compared to liraglutide.

The rate of any reported adverse events was 95.2% with semaglutide, 96.1% with liraglutide, and 95.3% with placebo. The number of reported serious side effects was higher with liraglutide (11%) compared to semaglutide (7.9%) or placebo (7.1%). Moreover, drug termination was more common in the liraglutide arm (12.6%) vs. semaglutide (3.2%) and placebo (3.5%). Gastrointestinal- and gallbladder-related symptoms were reported in 84.1% and 0.8% with semaglutide, 82.7% and 3.1% with liraglutide, and 55.3% and 1.2% with placebo. Hypoglycemia and acute pancreatitis were reported only with the liraglutide group (0.8% both). The injection site reactions were observed with liraglutide (11%) and placebo (5.9%) but not with semaglutide (0%).

In summary, the STEP 8 trial concluded that once-weekly semaglutide with lifestyle modifications was significantly superior to once-daily liraglutide with lifestyle modifications in body weight reduction and other cardiometabolic risk factors improvement.

## 9. Discussion

The STEP program demonstrated that once-weekly semaglutide with various intensity of lifestyle modifications was superior to placebo or once-daily liraglutide with lifestyle modifications in body weight reduction and other cardiometabolic risk factors improvement. The main secondary efficacy endpoints are summarized in Table 2. The STEP 2 trial included individuals with type 2 diabetes mellitus and obesity [23]. In the STEP 6 trial, only 25% of the patients had diabetes [27]. Conversely, the STEP 1, 3, 4, 5, and 8 trials did not include patients with type 2 diabetes mellitus [22,24,25,26,28], which may explain the superior weight loss in STEP 1, 3–6, and 8. The purpose of the STEP trials was for semaglutide 2.4 mg to gain regulatory approval and, as such, the two primary efficacy outcomes were percentage weight loss and the proportion of individuals achieving ≥5% weight loss at the endpoint.

In a recent meta-analysis of randomized controlled trials comparing the efficacy of different obesity medications, it showed that the percentage of bodyweight reduction from baseline with phentermine-topiramate was 7.97%, naltrexone-bupropion was 4.11%, orlistat was 3.16%, and liraglutide was 4.68%. Phentermine-topiramate and naltrexone-bupropion combinations were associated with the most adverse events. Their findings suggested that semaglutide might be the most effective among all the different obesity medications [15].

The STEP 4 trial (a withdrawal study) appraised the cessation of semaglutide therapy after a 20-week run-in interval. STEP 4 showed that individuals who continued semaglutide therapy had sustained a significant body weight loss contrasted with those who switched to placebo who started regaining weight [25]. Six of the STEP trials employed a semaglutide dose of 2.4 mg, except for the STEP 2 and STEP 6 trials, which also had an arm using semaglutide 1.0 mg and 1.7 mg, respectively. STEP 2 and 6 demonstrated that the higher semaglutide dose resulted in more body weight loss but also had fewer adverse effects, reflecting a dose–response effect. The STEP 5 trial had the longest duration of all STEP trials (104 weeks on medication) and explored the long-term effect of 2.4 mg semaglutide contrasted with placebo on body weight and various cardiometabolic risk factors over a two-year period. In the STEP 8 trial, the reduction in body weight was significantly greater with weekly semaglutide injection when compared to daily liraglutide injection, accompanied by significant improvements in various cardiometabolic risk factors. The analysis of exploratory secondary endpoints in all the STEP trials revealed beneficial effects on blood pressure, glycemic, lipid, inflammatory, and anthropometric parameters.

Overall, semaglutide had a good safety profile without any new safety signals not previously detected in other GLP-1 analogs. The rate of serious adverse events and proportion of side effects leading to drug termination was generally similar to other GLP-1 analogs. The tolerability profile and main adverse events for STEP 1–6 and 8 are presented in Table 3. The vast majority of the drug-associated adverse events were mild gastrointestinal-related symptoms. The rates of hypoglycemia, acute pancreatitis, gallbladder-related symptoms, and injection site reactions were low in the semaglutide groups and often comparable with the placebo groups. These will, however, need to be monitored in post-market-surveillance schemes.

The STEP trials have several strengths, including the scientifically robust methodologies, as reflected by the phase-III, large-sized, multicentric, double-blind, and placebo-controlled study designs. Limitations include the unintentional biased gender and ethnicity, as the vast majority of the recruited research participants were White females. Patients were recruited from routine clinical services where the usual demographic is reflected in the trials with a preponderance of females. These sociodemographic factors could have introduced a bias in the pooled outcomes. However, in the STEP 6 trial, all participants were east Asian, and the majority were males, and the outcomes were almost similar to the other STEP trials. Another limitation includes the short-term follow-up interval of roughly 68 weeks. This limitation was partly addressed in the STEP 5 trial, which provided a much longer follow-up of two years. However, obesity is a chronic disease and will require chronic treatment.

All STEP trials included a lifestyle intervention. However, only the STEP 3 trial incorporated very intensive lifestyle modifications, which included a low-calorie intake during the opening 8 weeks and then an additional 30 weeks of intensive behavioral therapy sessions with registered dieticians [24]. As a consequence, the patients in the placebo arm lost almost double the amount of weight recorded in the placebo arms of STEP 1, 2, 5, and 6 [22,23,26,27]. The placebo arm in STEP 3 lost 5.7% of weight, while the placebo arm in STEP 1 lost 2.4%, STEP 2 lost 3.4%, STEP 5 lost 2.6%, and STEP 6 lost 2.1%. Weight loss in the placebo arm of STEP 4 was 5%, but these patients were busy regaining weight after being treated with semaglutide for 20 weeks before being switched to only receiving standard lifestyle modifications. However, the approach of short-term drug treatment followed by standard lifestyle modifications in STEP 4 appeared as effective as very intensive lifestyle modification with placebo treatment. It was striking that the total weight loss achieved at the end of the treatment period in the semaglutide arms for STEP 1 was 14.9%, STEP 3 was 16%, STEP 4 was 17.4%, STEP 5 was 15.2%, STEP 6 was 13.2%, and STEP 8 was 15.8%. Only STEP 3 had very intensive lifestyle modifications, and it was expected that, similar to SCALE intensive behavior therapy, the addition of the intensive lifestyle modifications to semaglutide 2.4 mg would have added significantly more weight loss than when semaglutide 2.4 mg was combined with standard lifestyle modifications [29,30]. This raises the question of whether semaglutide 2.4 mg requires any lifestyle modification to be effective.

Other successful obesity treatments, such as bariatric surgery, do not appear to require lifestyle modifications to provide any more weight loss within the first year after surgery [31,32]. This may be due to the subject’s obesity being effectively treated with surgery. The changes in food intake behavior after successful obesity treatment may not be amplified by giving the patient a more stringent diet or exercise regimen. If this is true, then the cost of providing effective obesity care will significantly reduce in the first year because it is often the requirement for lifestyle modifications, which makes it difficult for practitioners to prescribe obesity treatments. This does not mean that lifestyle modifications may not be helpful in the longer term because, as the STEP trials have shown, there are also non-responders and partial responders to semaglutide. The addition of lifestyle modification may provide additional weight loss to those who only partially respond to the medication and thus result in substantial additional health gain; for example, in STEP 3 trial, 75.3% of patients achieved >10% weight loss compared to 69.1% in STEP 1.

Another major unexplored consequence of >15% weight loss with semaglutide may be the inevitable lean muscle mass loss. This is also evident after liraglutide and bariatric surgery [32,33,34,35]. The challenge is that patients who consume so few calories because of the effective medication cannot consume enough protein in their daily intake to stop them from becoming catabolic and losing muscle. Exercising these patients further may only result in them becoming more catabolic and losing even more muscle mass [36]. Thus, nutritional therapies once patients are in a steep negative energy balance may have to focus on optimizing protein intake to prevent muscle mass loss. This may further improve the functional gains made by patients if they can achieve 15% weight loss and maintain most of the lean muscle mass [31,37]. These hypotheses, however, require further testing, as our suggestions are purely speculation based on the similarities between trials, which used intensive or less intensive lifestyle changes.

## 10. Conclusions

In summary, the results of the STEP trials supported the efficacy of high-dose, once-weekly 2.4 mg semaglutide on body weight reduction among individuals with obesity. While semaglutide resulted in more gastrointestinal-related side effects, the medication appeared generally safe and well tolerated. The drug may be so effective that the role of nutritional therapy may have to be redefined, and a shift away from using nutritional therapy to achieve more weight loss to rather using nutritional therapy to achieve more health gain may be required.

## Figures and Tables

**Table 1 nutrients-14-02217-t001:** Mechanism of action and approval status of main obesity medications [15,16].

Medication	Mechanism of Action	Year of FDA Approval	Year of EMA Approval
Naltrexone-bupropion	Reduces energy consumption via potential synergistic effects on pro-opiomelanocortin neurons.	2014	2015
Phentermine-topiramate	Phentermine is an amphetamine-like appetite suppressant working through inhibition of noradrenaline reuptake in the hypothalamus, while topiramate is an anticonvulsant, which has some weight-loss effects, but its mechanism of action is not fully understood.	2012	Not approved
Orlistat	Decreases fat absorption by inhibiting the gastric and pancreatic lipases.	1999	1998
Setmelanotide	Melanocortin 4 (MC4) receptor agonist, works by restoring impaired MC4 receptor pathway activity caused by genetic deficits.	2020	2021
Liraglutide	Glucagon-like peptide-1 (GLP-1) receptor agonist that reduces hunger. Additionally, increases satiety.	2014	2015
Semaglutide	Glucagon-like peptide-1 (GLP-1) receptor agonist that reduces hunger. Additionally, increases satiety.	2021	2022

**Table 2 nutrients-14-02217-t002:** Change in the main efficacy endpoints from baseline to end of treatment for STEP 1–6 and 8 [22,23,24,25,26,27,28].

Parameter	STEP 1		STEP 2			STEP 3		STEP 4 ^b^		STEP 5		STEP 6			STEP 8	
	Semaglutide 2.4 mg	Placebo	Semaglutide 2.4 mg	Semaglutide 1 mg	Placebo	Semaglutide 2.4 mg	Placebo	Semaglutide 2.4 mg	Placebo	Semaglutide 2.4 mg	Placebo	Semaglutide 2.4 mg	Semaglutide 1.7 mg	Placebo	Semaglutide 2.4 mg	Liraglutide 3 mg
Body weight change (%)	–14.9	–2.4	–9.6	–7.0	–3.4	–16	–5.7	–7.9	6.9	–15.2	–2.6	–13.2	–9.6	–2.1	–15.8	–6.4
Participants with ≥5% weight loss (%)	86.4	31.5	68.8	57.1	28.5	86.6	47.6	88.7	47.6	77.1	34.4	83	72	21	87.2	58.1
Participants with ≥10% weight loss (%)	69.1	12.0	45.6	28.7	8.2	75.3	75.3	79.0	20.4	61.8	13.3	61	42	5	70.9	25.6
Participants with ≥15% weight loss (%)	50.5	4.9	25.8	13.7	3.2	55.8	13.2	63.7	9.2	52.1	7.0	41	24	3	55.6	12.0
WC (cm)	–13.54	–4.13	–9.4	–6.7	–4.5	–14.6	–6.3	−6.4	3.3	–14.4	–5.2	–11.1	–7.7	–1.8	–13.2	–6.6
BMI (kg/m^2^)	–5.54	–0.92	–3.5	–2.5	–1.3	–6.0	–2.2	−2.6	2.2	NA	NA	–4.3	–3.1	–0.6	NA	NA
CRP ^a^ (mg/dL)	0.47	0.85	0.51	0.58	0.83	–59.6	–22.9	NA	NA	–56.7	–7.8	0.39	0.64	0.92	–52.6	–24.5
SBP (mmHg)	–6.16	–1.06	–3.9	–2.9	–0.5	–6.3	–1.6	0.5	4.4	–5.7	–1.6	–11.0	–12.0	–5.0	–5.7	–2.9
DBP (mmHg)	–2.83	–0.42	–1.6	–0.6	–0.9	–3.0	–0.8	0.3	0.9	–4.5	–0.8	–5	–5	–3	–5.0	–0.5
HbA1c (%)	–0.45	–0.15	–1.6	–1.5	–0.4	–0.51	–0.27	−0.1	0.1	–0.43	–0.10	–1.0	–0.9	0.0	–0.2	–0.1
FPG (mg/dL)	–8.35	–0.48	–2.1	–1.8	–0.1	–6.73	–0.65	−0.8	6.7	–7.6	1.6	–19.3	–18.3	1.7	–8.3	–4.3
TC ^a^	0.97	1.00	0.99	0.98	0.99	–3.8	2.1	5	11	NA	NA	0.91	0.93	1.00	–7.1	–0.1
LDL ^a^	0.97	1.01	1.00	0.99	1.00	–4.7	2.6	1	8	NA	NA	0.86	0.90	0.95	–6.5	0.9
TG ^a^	0.78	0.93	0.78	0.83	0.91	–22.5	–6.5	−6	15	–19.0	3.7	0.79	0.78	1.05	–20.7	–11.0

^a^ The data were presented as ratios of end of treatment to baseline for STEP 1, 2, and 6 and change % for STEP 3–5 and 8; ^b^ Change from week 20 (run-in period) to end of treatment. Abbreviations: WC, waist circumference; BMI, body mass index; CRP, C-reactive protein; SBP, systolic blood pressure; DBP, diastolic blood pressure; HbA1c, hemoglobin A1c; FPG, fasting plasma glucose; TC, total cholesterol; LDL, low-density lipoprotein; TG, triglyceride; NA, not available.

**Table 3 nutrients-14-02217-t003:** Tolerability profile and main adverse events for STEP 1–6 and 8 [22,23,24,25,26,27,28].

Parameter, *n* (%)	STEP 1		STEP 2			STEP 3		STEP 4 ^a^		STEP 5		STEP 6			STEP 8	
	Semaglutide 2.4 mg	Placebo	Semaglutide 2.4 mg	Semaglutide 1 mg	Placebo	Semaglutide 2.4 mg	Placebo	Semaglutide 2.4 mg	Placebo	Semaglutide 2.4 mg	Placebo	Semaglutide 2.4 mg	Semaglutide 1.7 mg	Placebo	Semaglutide 2.4 mg	Liraglutide 3 mg
Any adverse event	1171 (89.7)	566 (86.4)	353 (87·6)	329 (81·8)	309 (76·9)	390 (95.8)	196 (96.1)	435 (81.3)	201 (75.0)	NA (96.1)	NA (89.5)	171 (86)	82 (82)	80 (79)	120 (95.2)	122 (96.1)
Serious adverse event	128 (9.8)	42 (6.4)	40 (9.9)	31 (7.70)	37 (9.2)	37 (9.1)	6 (2.9)	41 (7.7)	15 (5.6)	NA (7.9)	NA (11.8)	10 (5)	7 (7)	7 (7)	10 (7.9)	14 (11.0)
Adverse events leading to trial product discontinuation	92 (7.0)	20 (3.1)	25 (6.2)	20 (5.0)	14 (3.5)	24 (5.9)	24 (5.9)	13 (2.4)	6 (2.2)	NA (5.9)	NA (4.6)	5 (3)	3 (3)	0	4 (3.2)	16 (12.6)
Gastrointestinal disorders	969 (74.2)	314 (47.9)	256 (63.5)	231 (57.5)	138 (34.3)	337 (82.8)	129 (63.2)	224 (41.9)	70 (26.1)	125 (82.2)	82 (53.9)	118 (59)	64 (64)	30 (30)	106 (84.1)	105 (82.7)
Gallbladder-related disorders	34 (2.6)	8 (1.2)	1 (0.2)	4 (1.0)	3 (0.7)	20 (4.9)	3 (1.5)	15 (2.8)	10 (3.7)	4 (2.6)	2 (1.3)	2 (1)	1 (1)	1 (1)	1 (0.8)	4 (3.1)
Hypoglycemia	8 (0.6)	5 (0.8)	23 (5.7)	22 (5.5)	12 (3.0)	2 (0.5)	0	3 (0.6)	3 (1.1)	4 (2.6)	0	0	0	0	0	1 (0.8)
Acute pancreatitis	3 (0.2)	0	1 (0.2)	0	1 (0.2)	0	0	0	0	0	0	0	0	0	0	1 (0.8)
Injection site reactions	65 (5.0)	44 (6.7)	12 (3.0)	6 (1.5)	10 (2.5)	22 (5.4)	12 (5.9)	14 (2.6)	6 (2.2)	10 (6.6)	15 (9.9)	4 (2)	0	0	0	14 (11.0)
Diarrhoea	412 (31.5)	104 (15.9)	86 (21.3)	89 (22.1)	48 (11.9)	147 (36.1)	45 (22.1)	77 (14.4)	19 (7.1)	NA	NA	32 (16)	22 (22)	6 (6)	35 (27.8)	23 (18.1)
Constipation	306 (23.4)	62 (9.5)	70 (17.4)	51 (12.7)	22 (5.5)	150 (36.9)	50 (24.5)	62 (11.6)	17 (6.3)	NA	NA	52 (26)	19 (19)	3 (3)	49 (38.9)	40 (31.5)
Nausea	577 (44.2)	114 (17.4)	136 (33.7)	129 (32.1)	37 (9.2)	237 (58.2)	45 (22.1)	75 (14.0)	13 (4.9)	NA	NA	35 (18)	18 (18)	4 (4)	77 (61.1)	75 (59.1)
Vomiting	324 (24.8)	43 (6.6)	88 (21.8)	54 (13.4)	11 (2.7)	111 (27.3)	22 (10.8)	55 (10.3)	8 (3.0)	NA	NA	19 (9)	10 (10)	2 (2)	32 (25.4)	32 (25.4)
Nasopharyngitis	281 (21.5)	133 (20.3)	68 (16.9)	47 (11.7)	59 (14.7)	90 (22.1)	90 (22.1)	58 (10.8)	39 (14.6)	NA	NA	53 (27)	24 (24)	18 (18)	10 (7.9)	11 (8.7)

^a^ Data from week 20 (run-in period) to end of treatment.

## Data Availability

Not applicable.

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
