# Peer review of "The Impact Once-Weekly Semaglutide 2.4 mg Will Have on Clinical Practice: A Focus on the STEP Trials"

_nutrients, 2022, doi:10.3390/nu14112217_

Round 1

Reviewer 1 Report

Congratulations to the authors on a well-done study. The work is well-written and comprehensive. The work is also specific and sound in terms of methodology. At work, I don't have any revisions to do. It is acceptable to me in its current form.

Author Response

We thank the reviewer for the kind comment.

Reviewer 2 Report

Alabduljabbar and colleagues prepared a review regarding the STEP trials. Although by reading the title, it leads the reader to think the subject is about the impact of nutritional therapy in obesity, that was not the case. In fact, there are a few problems with the abstract, as it does not reflect the manuscript. For example, the paper is definitively not exploring "the impact of the behavioral programs on the effect of semaglutide 2.4mg on weight loss" (lines 18-19), but it is more likely basically summarizing seven STEP trials, including the inclusion/exclusion criteria, the set-up, and the results from such trials. Also by stating "semaglutide was associated with more gastrointestinal related side effects" (line 21-22), it should state compared to what.

The manuscript is very repetitive, making it exhausting to read, and the authors should think of a table to organize the data from the trials.

At the (maybe too short) introduction, bariatric surgery should be referred (and not only at the discussion).

It would also be important to add whether such drugs (lines 42-3) approved by FDA, were also approved by EMA (and in which years, respectively).

At introduction, the authors could indicate how many STEPS there were, also why more than one was developed, ie, which specific aims, and the years they were performed.

At STEP 1, it is not clear if the lifestyle intervention (line 69-70) was daily or weekly.

Please introduce what are the "physical function scores" (lines 82-3).

At the short summaries, at the end of STEP 1, 2 and 6, the authors conclude about the "patients with obesity", although patients with BMI below 30 kg/m2 were included, which might not be accurate to state like this.

The authors could speculate why the recruited patients were mainly females.

As a whole, the authors only briefly discuss the importance of nutrition, but at the trials this was not evaluated, as there was not a group of patients without the interventional lifestyle. Also, though the skeletal muscle mass loss referred by the authors, is a very relevant problem related to obesity remission, it seems the STEPs trials did not evaluate body composition, so it might be very misleading to refer to muscle mass and function at the abstract. 

Author Response

Comment 1. Alabduljabbar and colleagues prepared a review regarding the STEP trials. Although by reading the title, it leads the reader to think the subject is about the impact of nutritional therapy in obesity, that was not the case. In fact, there are a few problems with the abstract, as it does not reflect the manuscript. For example, the paper is definitively not exploring "the impact of the behavioral programs on the effect of semaglutide 2.4mg on weight loss" (lines 18-19), but it is more likely basically summarizing seven STEP trials, including the inclusion/exclusion criteria, the set-up, and the results from such trials.

Response 1. We thank the reviewer for the detailed review of our manuscript which we think will significantly improve the review. As requested we have adjusted the title to reflect the content of the review.

Comment 2. Also by stating "semaglutide was associated with more gastrointestinal related side effects" (line 21-22), it should state compared to what.

Response 2. As requested we have amended line 21-22 to clearly state that "semaglutide was associated with more gastrointestinal related side effects compared to placebo".

Comment 3. The manuscript is very repetitive, making it exhausting to read, and the authors should think of a table to organize the data from the trials.

Response 3. As requested we have added 2 tables summarizing the data.

Comment 4. At the (maybe too short) introduction, bariatric surgery should be referred (and not only at the discussion).

Response 4. We have now referred to bariatric surgery as one of the treatments for obesity on and added more information in the introduction.

Comment 5. It would also be important to add whether such drugs (lines 42-3) approved by FDA, were also approved by EMA (and in which years, respectively).

Response 5. We have added on page 2 line 39-41 that “ These medications, except phentermine-topiramate, are also approved by the European Medicines Agency (EMA) to be use in Europe.” The mechanism of actions and the approval status for these medications are summarized in Table 1.

Comment 6. At introduction, the authors could indicate how many STEPS there were, also why more than one was developed, ie, which specific aims, and the years they were performed.

Response 6. As requested, we have briefly indicated on page 2 that there are 15 STEP trials and we have explained the objectives of these trials and why there are more than one trial.

Comment 7. At STEP 1, it is not clear if the lifestyle intervention (line 69-70) was daily or weekly.

Response 7. We amended the line to state that “The protocol included an unsupervised lifestyle intervention administered to all participants, consisting of a daily 500-kcal deficit diet and weekly 150 min of physical activity.”

Comment 8. Please introduce what are the "physical function scores" (lines 82-3).

Response 8. As requested, we amended the line to state that physical function scores were assessed by the 36-item Short Form Health Survey (SF-36) and the Impact of Weight on Quality of Life–Lite Clinical Trials Version (IWQOL-Lite-CT) questionnaire.

Comment 9. At the short summaries, at the end of STEP 1, 2 and 6, the authors conclude about the "patients with obesity", although patients with BMI below 30 kg/m2 were included, which might not be accurate to state like this.

Response 9. We apologise for this mistake and we have amended the manuscript to use the phase “patients with a BMI > 27 kg/m2

Comment 10. The authors could speculate why the recruited patients were mainly females.

Response 10. We have added a comment in the discussion that patients were recruited from routine clinical services where the usual demographic is reflected in the trials with a preponderance of females.

Comment 11. As a whole, the authors only briefly discuss the importance of nutrition, but at the trials this was not evaluated, as there was not a group of patients without the interventional lifestyle.

Response 11. We agree with the reviewer and we have added a sentence to state that all the STEP trials included a lifestyle intervention. We have made clear that our suggestion that nutritional interventions should be re-evaluated are purely speculation based on the similarities between trials which used intensive or less intensive lifestyle changes.

Comment 12. Also, though the skeletal muscle mass loss referred by the authors, is a very relevant problem related to obesity remission, it seems the STEPs trials did not evaluate body composition, so it might be very misleading to refer to muscle mass and function at the abstract.

Response 12. We agree with the reviewer and we have deleted these speculative comments from the abstract.

Reviewer 3 Report

In this work Alabduljabbar and coworkers present the review of clinical trials for obesity management with the involvement of semaglutide intervention. The revision of the weight loss effect includes not only usage of semaglutide but also comments the influence of other separate lifestyle factors, i.e. caloric restrictions, physical activity. I am aware that submitted review is based on the semaglutide, however I suggest to present shortly the molecular mechanism of action of other medications mentioned in the introduction - this could present the reason for searching for the medicines that are more effective and have less harmful side effects, like semaglutide.

The short research allowed me to find “A quick guide to the STEP trials; A comparable work has been published by Eleanor McDermid in 02-11-2021 | Semaglutide | At a glance | Article;  which has been updated in February 2022” https://diabetes.medicinematters.com/semaglutide/obesity/quick-guide-step-trials/18854832; however the Authors did not include this work in their manuscript as the reference.

In the introduction I suggest to present the number of STEPS (8) with the information/explanation about STEP7, since it has not been mentioned in the work.

Since it is a review, please add the data to the key words/parameters presented in lines 78-92.

Line 102 – it is not presented clearly if the dose 1.0mg of semaglutide was used simultaneously wit 2.4 mg and placebo, or was it a step before reaching the 2.4 mg dose.  Please add the data to the key words/parameters presented in lines 114-120.

Lines 160-161 – the sentence “the treatment (semaglutide with very intensive behaviour therapy or placebo with very intensive behaviour therapy) lasted for 68 weeks” is repeated, since it was presented in lines 142-143. Therefore I suggest to modify the text “The duration of the study was 75 weeks; the treatment (semaglutide with very intensive behaviour therapy or placebo with very intensive behaviour therapy) lasted for 68 weeks, trailed by a follow-up period of seven weeks with no medication.”. Please add the data to the key words/parameters presented in lines 171-176.

Please add the data to the key words/parameters presented in lines 208-210 after 20 and 68-week intervention.

Please add the data to the key words/parameters presented in lines 295-298 and 333-335.

Since it is a review, please, present the differences between liraglutide and semaglutide which were used in STEP8.

In Discussion section  I suggest strongly to add the Table presenting the main results/achievements of STEP1-8 – this will allow to compare the trials more clearly.  

In line 410-411 the Authors wrote “This raises the question whether semaglutide 2.4mg requires any lifestyle modification to be effective.”. Can the Authors present more detailed information about diet that was used for “low-calorie intake” behaviour in all corresponding STEPs? The same with “standard lifestyle modifications” and “ very intensive lifestyle modification” - the dietary patterns (type of food ingredients) and type of physical activity may influence the weight loss. Can the Authors compare these results with other medications (without  bariatric surgery) used for obesity treatment?

In summary I suggest the minor revision.

Author Response

Comment 1. In this work Alabduljabbar and coworkers present the review of clinical trials for obesity management with the involvement of semaglutide intervention. The revision of the weight loss effect includes not only usage of semaglutide but also comments the influence of other separate lifestyle factors, i.e. caloric restrictions, physical activity. I am aware that submitted review is based on the semaglutide, however I suggest to present shortly the molecular mechanism of action of other medications mentioned in the introduction - this could present the reason for searching for the medicines that are more effective and have less harmful side effects, like semaglutide.

Response 1. We thank the reviewer for the detailed assessment of our manuscript and the helpful comments. As requested we have added a table summarizing the molecular mechanism of action of other medications mentioned in the introduction.

Comment 2. The short research allowed me to find “A quick guide to the STEP trials; A comparable work has been published by Eleanor McDermid in 02-11-2021 | Semaglutide | At a glance | Article;  which has been updated in February 2022” https://diabetes.medicinematters.com/semaglutide/obesity/quick-guide-step-trials/18854832; however the Authors did not include this work in their manuscript as the reference.

Response 2. We thank the reviewer for the comment. We have looked into the useful webpage. It provides 1 to 2 paragraphs high level summary and a link to the clinicaltrial.gov website. As this is not a peer reviewed academic article we would appreciate if the reviewer will allow us not to add it to our references.

Comment 3. In the introduction I suggest to present the number of STEPS (8) with the information/explanation about STEP7, since it has not been mentioned in the work.

Response 3. We have stated that “Six of the program trials (STEP 1-4, 6 and 8) were published, the STEP 5 trial has been completed but not yet published and the remining trials, including STEP 7, have not been completed yet.”

Comment 4. Since it is a review, please add the data to the key words/parameters presented in lines 78-92.

Response 4. We have added this on table 2.

Comment 5. Line 102 – it is not presented clearly if the dose 1.0mg of semaglutide was used simultaneously with 2.4 mg and placebo, or was it a step before reaching the 2.4 mg dose. 

Response 5. We have clarified that in most of the STEP trials patients progressed to semaglutide 2.4mg and only used semaglutide 1mg as part of the dose titration. The difference in STEP 2 where semaglutide 2.4mg was directly compared with patients on semaglutide 1mg for the duration of the trial.

Comment 6. Please add the data to the key words/parameters presented in lines 114-120.

Response 6. We have added this on table 2.

Comment 7. Lines 160-161 – the sentence “the treatment (semaglutide with very intensive behaviour therapy or placebo with very intensive behaviour therapy) lasted for 68 weeks” is repeated, since it was presented in lines 142-143. Therefore I suggest to modify the text “The duration of the study was 75 weeks; the treatment (semaglutide with very intensive behaviour therapy or placebo with very intensive behaviour therapy) lasted for 68 weeks, trailed by a follow-up period of seven weeks with no medication.”.

Response 7. We have made the changes as suggested by the reviewer.

Comment 8. Please add the data to the key words/parameters presented in lines 171-176.

Response 8. We have added this on table 2.

Comment 9. Please add the data to the key words/parameters presented in lines 208-210 after 20 and 68-week intervention.

Response 9. We have added this on table 2.

Comment 10. Please add the data to the key words/parameters presented in lines 295-298 and 333-335.

Response 10. We have added this on table 2.

Comment 11. Since it is a review, please, present the differences between liraglutide and semaglutide which were used in STEP8.

Response 11. As requested we added a statement to briefly explain the difference between liraglutide and semaglutide which were used in STEP8: “Both semaglutide and liraglutide are long-acting GLP-1 analogues. As a result of the substitution of amino acids that prevents the degeneration of dipeptidyl peptidase 4 and addition of C18 fatty acids, Semaglutide has a half-life of 165 hours whereas liraglutide’s half-life is about 13 hours”.

Comment 12. In Discussion section I suggest strongly to add the Table presenting the main results/achievements of STEP1-8 – this will allow to compare the trials more clearly. 

Response 12. As requested, we have added 2 tables summarizing the data.

Comment 13. In line 410-411 the Authors wrote “This raises the question whether semaglutide 2.4mg requires any lifestyle modification to be effective.”. Can the Authors present more detailed information about diet that was used for “low-calorie intake” behaviour in all corresponding STEPs? The same with “standard lifestyle modifications” and “ very intensive lifestyle modification” - the dietary patterns (type of food ingredients) and type of physical activity may influence the weight loss.

Response 13. We thank the reviewer for his suggestion. Unfortunately, the STEP1, 2, 4, 5, 6, and 8 trials reported a general diet with 500kcal deficit and physical activity of 150 min weekly, without much detail provided.  Detailed information of the lifestyle intervention is however provided for the STEP 3 trial and we have described this on page 5 Paragraph 3-4.

Comment 14. Can the Authors compare these results with other medications (without bariatric surgery) used for obesity treatment?

Response 14. We have briefly mentioned summary results from a meta-analysis of

randomised controlled trials on orlistat, liraglutide, naltrexone/buproprion combination, and phentamine/topiramate combination in Discussion section.